# Efficient Pyrolysis of Low-Density Polyethylene for Regulatable Oil and Gas Products by ZSM-5, HY and MCM-41 Catalysts

**Ting Liu [1], Yincui Li [1], Yifan Zhou [2], Shengnan Deng [1] and Huawei Zhang [1],***

[1] School of Environmental and Municipal Engineering, Qingdao University of Technology, Qingdao 266033, China

[2] College of Chemical and Biological Engineering, Shandong University of Science and Technology, Qingdao 266590, China

* Correspondence: sdkdzhw@163.com; Tel.: +86-0532-85071133

**Abstract:** In this research, catalytic cracking of low-density polyethylene (LDPE) has been carried out in the presence of three kinds of typical molecular sieves, including ZSM-5, HY and MCM-41, respectively. The effects of different catalysts on the composition and quantity of pyrolysis products consisting of gas, oil and solid material were systematically investigated and summarized. Specially, the three kinds of catalysts were added into LDPE for pyrolysis to obtain regulatable oil and gas products ($H_2$, $CH_4$ and a mixture of $C_2$–$C_4^+$ gaseous hydrocarbons). These catalysts were characterized with BET, $NH_3$-TPD, SEM and TEM. The results show that the addition of MCM-41 improved the oil yield, indicating that the secondary cracking of intermediate species in primary pyrolysis decreased with the case of the catalyst. The highest selectivity of MCM-41 to liquid oil (78.4% at 650 °C) may be attributed to its moderate total acidity and relatively high BET surface area. The ZSM-5 and HY were found to produce a great amount of gas products (61.4% and 67.1% at 650 °C). In particular, the aromatic yield of oil production reached the maximum (65.9% at 500 °C) when the ZSM-5 was used. Accordingly, with the three kinds of catalysts, a new environment-friendly and efficient recovery approach may be developed to obtain regulatable and valuable products by pyrolysis of LDPE-type plastic wastes.

**Keywords:** LDPE; ZSM-5; HY; MCM-41; catalytic pyrolysis

## 1. Introduction

Global demand for plastics is growing rapidly due to their widespread applications in many fields [1–3]. Plastic production has increased 20-fold over the past half century and is expected to exceed 500 million tons by 2050 [4]. There are many kinds of plastics, while polyethylene (PE) ranks first with 32%. Low-density polyethylene (LDPE) is one of the most widely applied plastic [5]. LDPE has a high degree of short- and long-chain branching, which prevents the chains from entering the crystal structure [4,6]. The environment and global ecosystems are negatively affected by the excessive use, improper management and disposal of plastics [3]. Therefore, how to achieve a clean and efficient utilization of waste plastics with high value, for example, regulatable oil and gas products, has become an urgent problem to be solved.

Pyrolysis is considered an emerging recycling technology that has attracted wide attention. It is the process of breaking polymer molecular chains and converting them into liquid oil, char and gases at a high temperature (300–900 °C) in an inert atmosphere [7,8]. Given the feasibility of regulatable gas, recycling waste plastics using pyrolysis is considered a promising treatment [9–11]. However, the factors influencing the pyrolysis process, such as temperature, pressure, catalyst type, heating mode, etc., are complex and there is still a great difference in the yield of gas products, restraining the application. At the same time, pyrolysis suffers from a large variety of reaction products, high pyrolysis temperatures and low yields of valuable chemicals. However, the introduction of catalysts can decrease

the reaction temperature as well as the activation energy of pyrolysis and change the way of plastic pyrolysis to achieve the selective collection of target products. At the same time, catalytic pyrolysis can promote the fracture of long-chain molecules to achieve light pyrolysis products [12] and reduce the viscosity of the liquid phase of pyrolysis, obtaining valuable chemical production.

There are various catalysts applied in the process of plastic pyrolysis, but the most widely employed catalysts are ZSM-5, Y-zeolite, FCC(fluid catalytic cracking) and MCM-41 [7]. Especially zeolites have found widespread applications in plastic cracking because of their structural advantages such as diverse skeleton structure, highly ordered pores, sufficient acid sites, large specific surface area as well as great stability [7,11].

ZSM-5 has been generally employed in the thermal cracking of waste plastic and gas adsorption-separation industry because of its strong acidity and shape selectivity. Ding et al. [13] and Du et al. [14] found that ZSM-5 is a kind of crystalline aluminosilicate material with a unique two-dimensional pore structure. The pores intersect each other with a diameter of 0.55 nm, which is conducive to the generation of hydrocarbons with a carbon number of less than 10. It also has excellent thermal stability and hydrothermal stability, strong acid resistance and anti-carbon deposition, adjustable acidity, great shape selectivity, isomerization capacity and other catalytic properties. Wei et al. [15] found that HY zeolite shows good catalytic performance with the advantages of regular pore structure, high stability as well as reactivity. At the same time, Ding et al. [16] found that HY is used as a catalyst for co-pyrolysis with LDPE, increasing from 23.5% to 80.4% as the ratio of HY to LDPE rose from 0 to 1:5. It is known that oil production and quality achieve the best balance at the HY to LDPE ratio of 1:10. Zhang et al. [2] reported that MCM-41 is a mesoporous material with a high surface area, which can enhance the yield of hydrocarbons and the quality of pyrolytic oil. Chi et al. [17] found MCM-41 has a unique advantage because its larger pore size makes macromolecular catalysis, adsorption and separation possible, reducing the diffusion resistance of molecules in the channel. Furthermore, MCM-41 has a high specific surface (about 1000 m$^2$/g), which provides adequate surface sites for adsorption and catalytic reactions of active ingredients. It also gets relatively fewer coke products. However, there have been few studies on the co-pyrolysis of different molecular sieves with LDPE [10,18–20] and the systematic analysis of the catalytic mechanism has not been perfected, lacking systematic analysis and summary of the pyrolysis characteristics of different molecular sieves and waste plastics.

In this paper, we aimed to make clean and efficient utilization of waste plastics with high value, obtaining regulatable gaseous products or liquids. Using ZSM-5, HY and MCM-41 as catalysts, catalytic pyrolysis of LDPE was performed in a fixed-bed reactor to achieve the three-phase products. The effect of pyrolysis temperature and type of catalysts on the product yield was explored. Furthermore, the characteristic and distribution of the pyrolysis products catalyzed by three kinds of catalysts were compared to obtain the interaction path and scheme of the catalytic pyrolysis. It provides the theoretical basis for the clean application of waste plastics, selectivity of valuable chemicals and selection of catalysts.

## 2. Results and Discussion

### 2.1. BET Results

Table 1 shows the structural properties of the three kinds of catalysts. MCM-41 has the largest BET-specific surface area caused by mesopores, facilitating multiple contacts of plastics with catalytic active centers and favoring the passage of large pyrolysis products (like olefins and aromatics) [21,22]. Compared to ZSM-5 and HY, MCM-41 obtains a larger average pore size (3.653 nm), specific surface area (962 m$^2$/g) and the total pore volume (0.718 cm$^3$/g), which is mainly manifested in the catalytic activity of MCM-41. ZSM-5 obtains the minimum average pore size of 0.411 nm, allowing the heavy chemicals to crack further.

**Table 1.** Textural properties of different catalysts.

| Catalysts | BET Surface Area (m²/g) | Total Pore Volume (cm³/g) | Average Pore Diameter (nm) |
|---|---|---|---|
| ZSM-5 | 361 | 0.206 | 0.411 |
| HY | 701 | 0.392 | 0.811 |
| MCM-41 | 962 | 0.718 | 3.653 |

## 2.2. Acid Properties of Zeolites

Figure 1 shows the NH₃-TPD results of the three kinds of catalysts. The weak, medium and strong acid sites of the catalysts correspond to the characteristic peaks at 155 °C, 275 °C and 505 °C, respectively. The results present that most of the acid sites of ZSM-5 are as the same as that of the HY, achieving uniform acid strength, while the MCM-41 zeolite-based catalyst with a $SiO_2/Al_2O_3$ ratio of 30 has low acid strength and no strong acid. The acidity shown in Table 2 further confirms the results obtained.

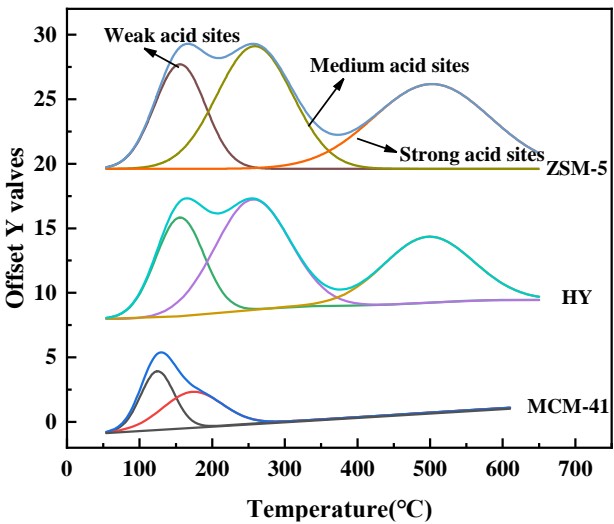

**Figure 1.** NH₃ adsorption/desorption isotherm distribution of samples.

**Table 2.** Acidity distribution of three kinds of catalysts.

| Catalysts | Acid Content (μmol/g) | | | |
|---|---|---|---|---|
| | Weak Acidity | Medium Acidity | Strong Acidity | Total Acidity |
| ZSM-5 | 368 | 838 | 549 | 1755 |
| HY | 366 | 748 | 652 | 1766 |
| MCM-41 | 138 | 221 | - | 359 |

In addition, the acid distribution of the catalyst was estimated by Gaussian fitting. As presented in Table 2, HY has the highest total acid contents and a strong acid site, which were 1766 μmol/g and 652 μmol/g, respectively. The MCM-41 has the lowest total acid content (359 μmol/g). The acidity of the catalysts has a great influence on the catalytic performance of the final product of plastic pyrolysis, which is covered in detail in Sections 2.4–2.6.

## 2.3. SEM and TEM Results

The SEM and TEM images of ZSM-5, MCM-41 and HY are exhibited in Figure 2. The ZSM-5 is constituted by clear quadrangular prism-like crystallites [23], which is consistent with the study reported by Haswin Kaur Gurdeep Singh et al. [24], with sizes ranging from 250–450 nm (Figure 2(a1)). And the image for ZSM-5 shows a relatively irregular sheet structure (Figure 2(a2)). It is worth noting that, in the presence of MCM-41, various

particles are uniformly distributed on the surface of the carrier (Figure 2(b2)), ranging from 20 nm to 90 nm, which might be conducive to its relatively higher specific surface area among three kinds of catalysts. As seen from the SEM image (Figure 2(c1)), the hierarchical HY zeolite retains its intact crystal structures [25,26], ranging from 50 nm to 80 nm and the regular sheet structure is observed in the HY (Figure 2(c2)).

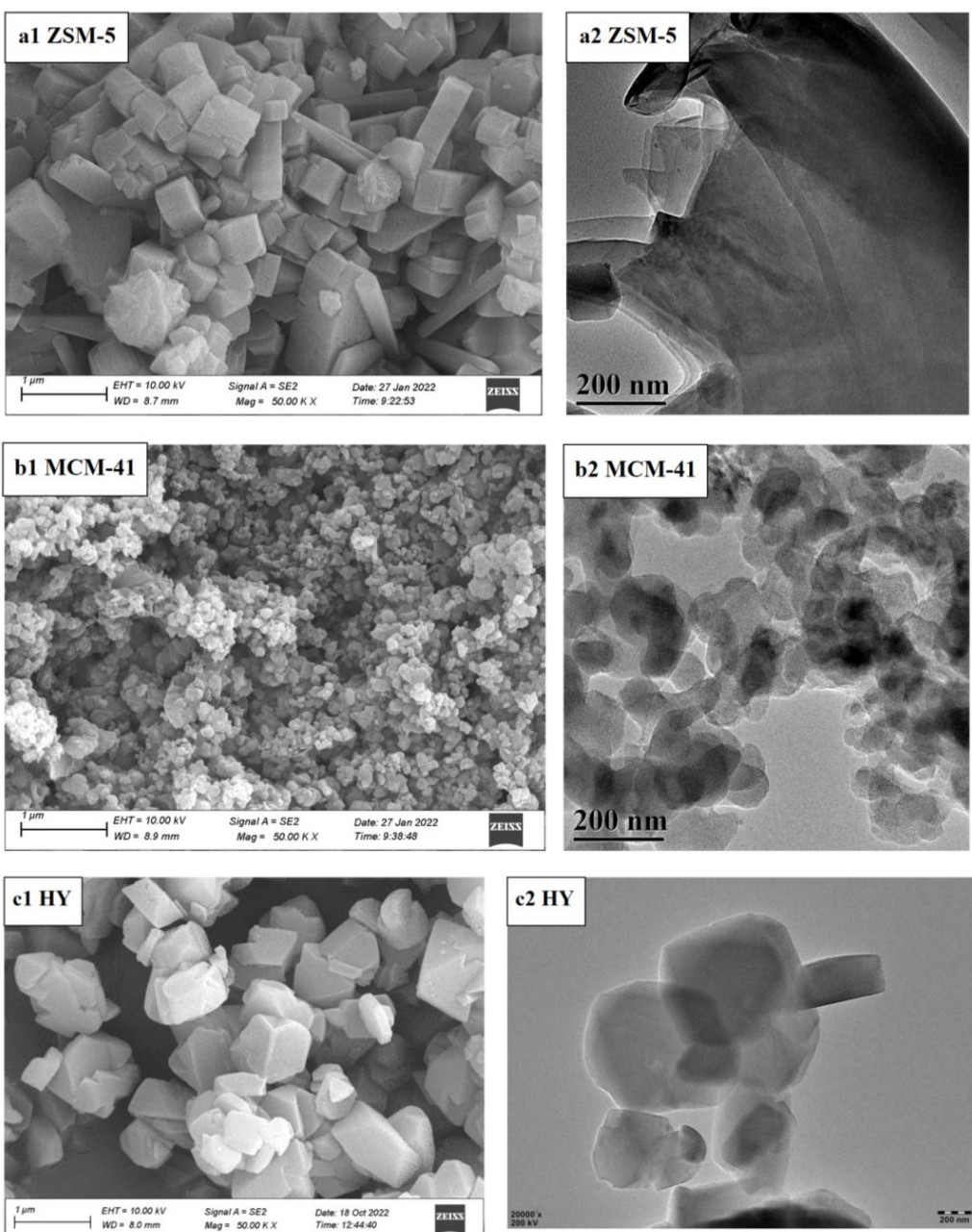

**Figure 2.** The SEM and TEM images of three kinds of catalysts. (**a1,b1,c1**): SEM; (**a2,b2,c2**): TEM.

### 2.4. Effects of Temperature on Gas–Liquid–Solid Three-Phase Yield of LDPE Pyrolysis

As depicted in Figure 3, the performance on catalytic conversion of LDPE over different catalysts from 450 °C to 650 °C was contrasted. It is obvious that the yield distribution was significantly affected by pyrolysis temperature. With the increase of pyrolysis temperature, the total yield of gas and oil enhanced largely while the yield of solid decreased greatly, which may be due to the decomposition and secondary reaction of LDPE pyrolysis volatiles [27]. As the temperature continued to rise, it provided more heat to the polymer, weakening the chain structure and causing more polymer chains to break [28] and the

trend is consistent with most research on polymer pyrolysis [29,30]. As can be seen from Figure 3a, the liquid phase yield was lower at 450 °C (31.3%) and at 500 °C (55.6%), with the increase in temperature, the conversion of the polymer improved [28], so the liquid yield rose to 82.0% at 550 °C. This is due to further increases in temperature causing further cracking of the oligomer to form smaller hydrocarbons in the form of gaseous compounds, while liquid production does not change significantly.

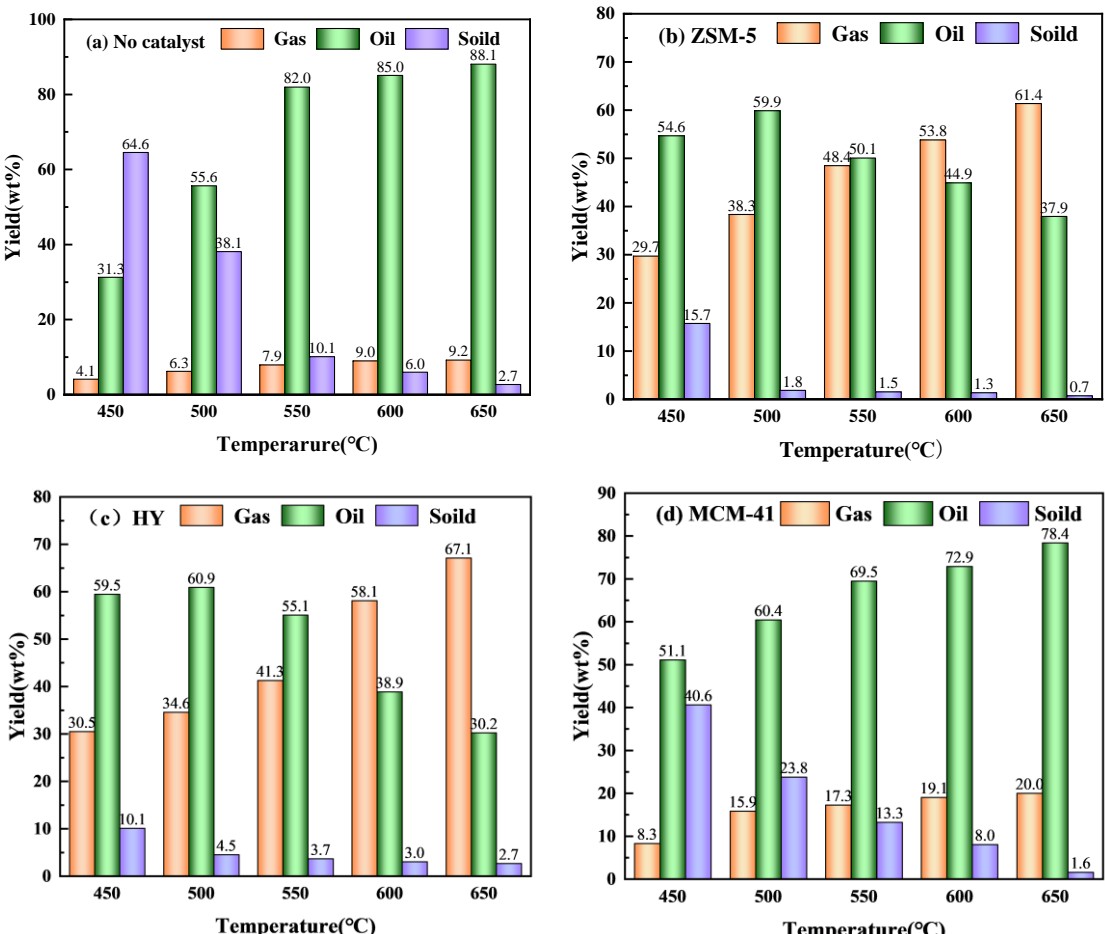

**Figure 3.** Three-phase yield diagram of LDPE (**a**): no catalyst; (**b**): ZSM-5 catalyst; (**c**): HY catalyst; (**d**): MCM-41 catalyst.

Compared to the case without catalysts, the pyrolysis gas yield of LDPE increased, indicating that the catalyst has a moderate acidity, leading to obvious secondary cracking of liquid oil [21,28]. As illustrated in Figure 3a, the liquid phase yield of non-catalytic pyrolysis at 500 °C occupied about 55.6%, which is similar to that of catalytic pyrolysis at 450 °C in Figure 3b. It can be concluded that the catalysts could significantly decrease the reaction temperature.

Furthermore, it can be observed that the ZSM-5 and HY catalysts resulted in much higher gas yields with increasing temperature while the MCM-41 obtained more oil yield. As seen from Figure 3b,c, there was no significant difference in gas yield between ZSM-5 and HY, both of which had higher gas yield than MCM-41. The yield of gas on the ZSM-5 catalyst increased from 29.7% to 61.4% as the temperature increased from 450 °C to 650 °C; however, the yield of oil on MCM-41 increased from 51.1% to 78.4%.

For ZSM-5, the main cause of such phenomenon comes from the function of the acidic sites and framework structure of ZSM-5 [31]. It was found that the interaction during catalytic pyrolysis could promote the formation of light molecular gases from chain-breaking volatiles [32]. At the same time, ZSM-5 has a smaller pore size and a larger

intracrystalline pore structure, allowing further cracking of heavy chemicals. Since the initial decomposition sample on the outer surface of the ZSM-5 can diffuse into the inner cavity of the ZSM-5, further decomposition into gaseous products resulted in very high gas yields [33]. Compared with MCM-41, the HY catalyst clearly provided a higher gas yield as a result of the strong acid sites and high acid density of the HY zeolite, which provided higher cracking activity than MCM-41 with only weak acid sites [34]. Additionally, the difference in gas yield among the three kinds of catalysts is due to the difference in carbon deposition [20,34,35], resulting in the difference in strong acid sites.

For MCM-41, it exhibited the greatest amounts of oil yield (78.4%) and the lowest amounts of gas yield (20.0%) at 650 °C. HY showed the second-highest oil yield (60.9%) and ZSM-5 presented a slightly lower oil yield (59.9%). This manifested that, in the case of these catalysts, secondary cracking was slightly enhanced and the difference in pyrolysis yields was largely as a result of the differences in acidity and structural properties discussed earlier. In addition, MCM-41 with uniform morphology was easy to produce the pyrolysis product with similar carbon distribution, leading to more oil products produced by MCM-41 than other catalysts [21].

### 2.5. Effects of Catalyst on the Composition and Quality of Gaseous Products

The gaseous product composition for non-catalytic and catalytic experiments from 450 °C to 650 °C are depicted in Figure 4. The pyrolysis gas consists of $H_2$, $CH_4$ and $C_2$–$C_4^+$ gaseous hydrocarbon mixtures. The contents of $H_2$, $CH_4$, $C_2$ and $C_3$ increased by 2.35%, 1.45%, 19.21% and 19.72% as the temperature rose from 450 °C to 650 °C, while $C_4^+$ gas reduced by 35.59%, which was mainly due to the $C_4^+$ gaseous products being further cleaved to $CH_4$ and other small molecule gases as temperature increased [27,28].

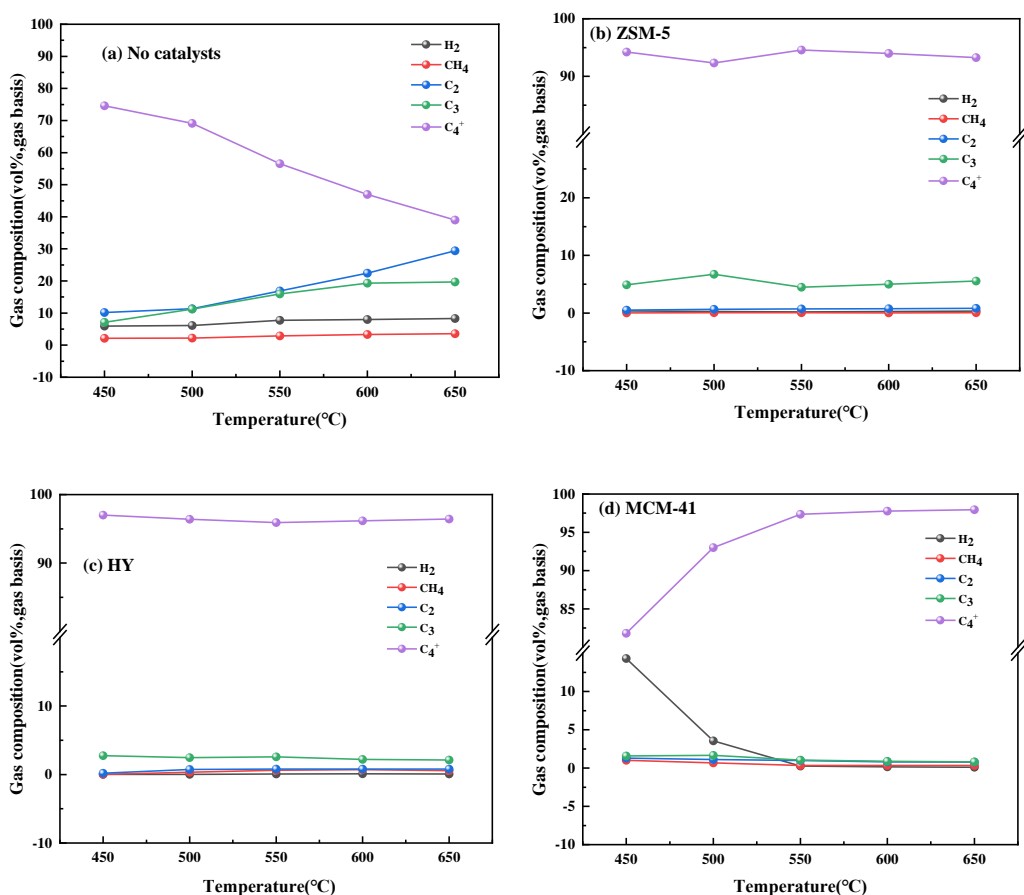

**Figure 4.** Comparison of gas−phase composition during LDPE pyrolysis with three kinds of catalysts (**a**) no catalysts; (**b**) ZSM-5; (**c**) HY; (**d**) MCM−41.

In the non-catalytic run, a considerable amount of $C_2$ and $C_3$ were observed. After adding the catalysts into pyrolysis, the number of $C_4^+$ gaseous products raised significantly. It is interesting that all the gaseous products of the catalyzed reactions showed a percentage of $C_4^+$ around 95% and MCM-41 at 500 °C. Additionally, the wider pore size distribution observed in HY and MCM-41 zeolite resulted in the diffusion of the reactant and product, which was more favorable for macromolecular hydrocarbons to enter the pore size of the molecular sieve to react with the recombinant gas [36]. As illustrated in Figure 4d, at temperatures of 450–500 °C, the gas phase products of MCM-41 had a higher content of $H_2$ and lower content of $C_4^+$ compared to both ZSM-5 and HY catalysts. This is because MCM-41 has a larger pore size and higher selectivity to heavier components. However, due to its weaker acidity, fewer active sites, lower catalytic activity and selectivity at lower temperatures, the content of hydrogen and methane was higher. However, as temperatures rose further, rapid product formation did not allow more cracking gas to occur in the reactor, so the heavier hydrocarbon component of the product increased, resulting in the formation of heavy hydrocarbons and low hydrogen [37]. Therefore, the main component of the gas phase product was $C_4^+$. The difference in the pyrolysis yield can be directly related to changes in the structural and acid properties of the catalysts.

### 2.6. Effects of Catalyst on Oil Distribution and Quantity

The carbon number distribution of oil is exhibited in Figure 5 and the major constituents of the oil product as well as relative content are presented in the Supplementary Material. As shown in Figure 5, in the experiment without catalysts, the oil products comprised four comparable fractions (<$C_{11}$, $C_{12}$–$C_{18}$, $C_{19}$–$C_{30}$ and aromatics), indicating that the carbon number distribution was relatively uniform compared to the catalytic experiment. As the temperature increased from 500 °C to 650 °C, the yield of light hydrocarbon fractions improved and the yield of heavy hydrocarbon fractions decreased in the non-catalytic run (Figure 5a). The liquid fractions are mainly composed of linear paraffins ($C_{10}$~$C_{30}$) and produced almost no aromatic hydrocarbons [3] (Tables S1 and S2). The pyrolysis of LDPE was the result of its characteristic long-carbon chain structure, converting the feedstock into wax rather than liquid oil [3].

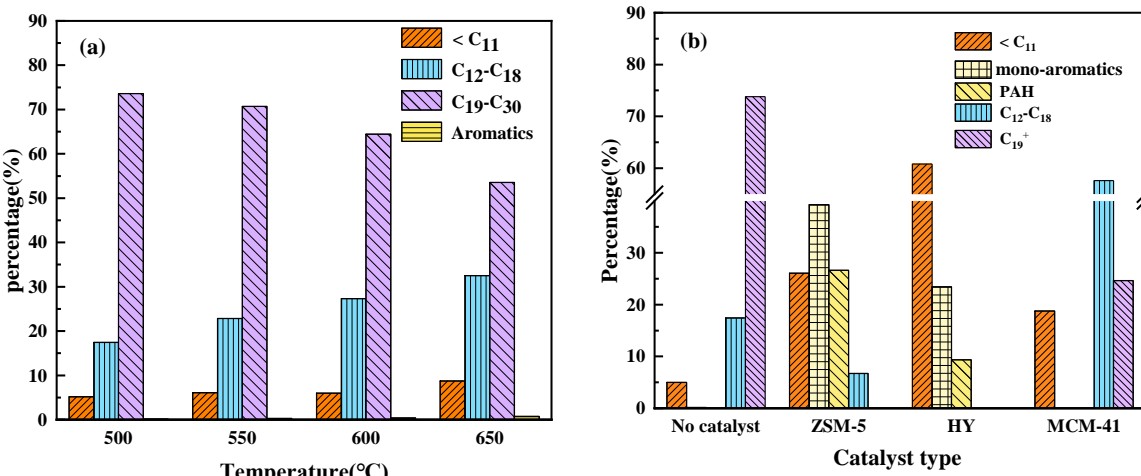

**Figure 5.** Distribution of the oil products in terms of carbon number (**a**) Liquid product composition of LDPE at different temperatures; (**b**) Liquid product composition of LDPE at different catalysts.

Moreover, it is obvious that the representative of diesel products is $C_{12}$–$C_{18}$ hydrocarbons and the MCM-41 catalyst has a potential application value in the use of plastic waste to produce diesel. Among the three kinds of catalysts, ZSM-5 presented the especially high selectivity for the aromatics and low selectivity for the $C_{12}$–$C_{18}$ fraction. Because ZSM-5 exhibited the second-highest total acid site, it is not difficult to infer that high diesel production was due, in part, to the mild acidity as well as excessive cracking of hydrocarbons.

Compared to the pyrolysis of LDPE without catalysts, the contents of mono-aromatics and polycyclic aromatic hydrocarbons (PAHs) in ZSM-5 catalytic pyrolysis were higher by 65.9% (Figure 5b). The pore size and structure of ZSM-5 played an important role in the formation of aromatic hydrocarbons, owing to its shape selectivity [20,38]. It is well-known that ZSM-5 has acid sites, a suitable pore size and shape selectivity, which is beneficial for the formation of aromatics [39] and the conversion of aliphatics to aromatic production using the Diels–Alder reaction [40]. It is worth noting that MCM-41 displayed the lowest selectivity for the aromatic compounds due to its weak aromatizing ability [19].

Furthermore, the results showed that the alkane content of heavy hydrocarbon fractions obtained during the catalytic process was more than the olefin content (Tables S3–S5). This is consistent with the conclusion in the study of Liu et al. [41], which may be caused by alkylation of the primary intermediate.

### 2.7. Effect of Catalysts on Some Reaction Pathways

It is concluded that the pyrolysis of LDPE followed the random-chain-breaking mechanism and the catalytic thermal decomposition of LDPE underwent the carbocation theory [4,8,33]. The catalytic effect of the catalysts was primarily due to their acidity during pyrolysis. The carbonate ion theory was based on the acid sites of the catalyst [33].

Thermal cracking of LDPE is often related to the free-radical mechanism [28]. The thermal pyrolysis of LDPE was partly caused by the homolysis of C–C bonds in the polymer chain under thermal action. As shown in Figure 6, LDPE formed free radical fragments of different lengths through random fracture (Step 1 and Step 2). Then the terminal free radical fragments generated alkenes through a hydrogen transfer reaction and further bond-breaking reaction (Step 3) and alkanes and hydrogen gas were further generated through a bimolecular reaction (Step 4).

As shown in Figure 7, the scheme of catalyst participation in LPDE pyrolysis reaction pathways is proposed. When the ZSM-5 was introduced into the pyrolysis process, the smaller pore size and larger pore structure of ZSM-5 allowed the initial decomposition sample on the outer surface to diffuse into the interior of ZSM-5, favoring the recombinant fraction to further decompose into gaseous products, resulting in higher gas production. Additionally, there was a higher aromatic content in the liquid phase of the ZSM-5 catalyst, probably due to its high acidity and shape selectivity. The acid sites could facilitate the formation of aromatics by catalyzing hydrogen transfer reactions and the Diels–Alder reactions [42] and the gas–liquid products underwent further aromatization. The hydrogen transfer reaction is considered to be the main source of aromatics and alkanes. Dehydrogenation active sites can promote the Diels–Alder reactions and cyclization intermediates. In addition, the heavy aromatic hydrocarbons were more easily decomposed into light aromatic hydrocarbons by hydrogenation than monocyclic and bicyclic aromatic hydrocarbons.

The wider pore size distribution observed in HY zeolite contributed to the diffusion of the reactant and product [36]. Therefore, the main component of the gas phase product was $C_4^+$, while the content of recombinant fraction ($C_{19}^+$) in the liquid phase product was seldom. Due to the strong acid site and high acid density of HY, it provided more pyrolysis activity leading to a higher gas yield. Bimolecular pyrolysis and hydrogen transfer reaction on HY can produce a large number of incondensable gas and a high yield of alkenes and alkanes below $C_{11}$.

When the MCM-41 was added to the pyrolysis, its mesoporous structure could greatly promote the accessibility of macromolecules to zeolite, reduce residence time, inhibit the secondary reaction and thus improve the yield of liquid [25]. Meanwhile, the mesoporous catalyst has a large pore volume and pore size, which was conducive to the free diffusion of the primary thermal decomposition molecules of LDPE and was easy to be converted into liquid products [43]. Therefore, compared with ZSM-5 and HY, MCM-41 had a greater promotion effect on crude oil fractions, which was less favorable for cracking and aromatization reactions, resulting in a higher content of $C_{12}$–$C_{18}$. The lower proportion of

aromatics obtained by MCM-41 may be related to its lower acid strength, lower catalytic activity and poor selectivity.

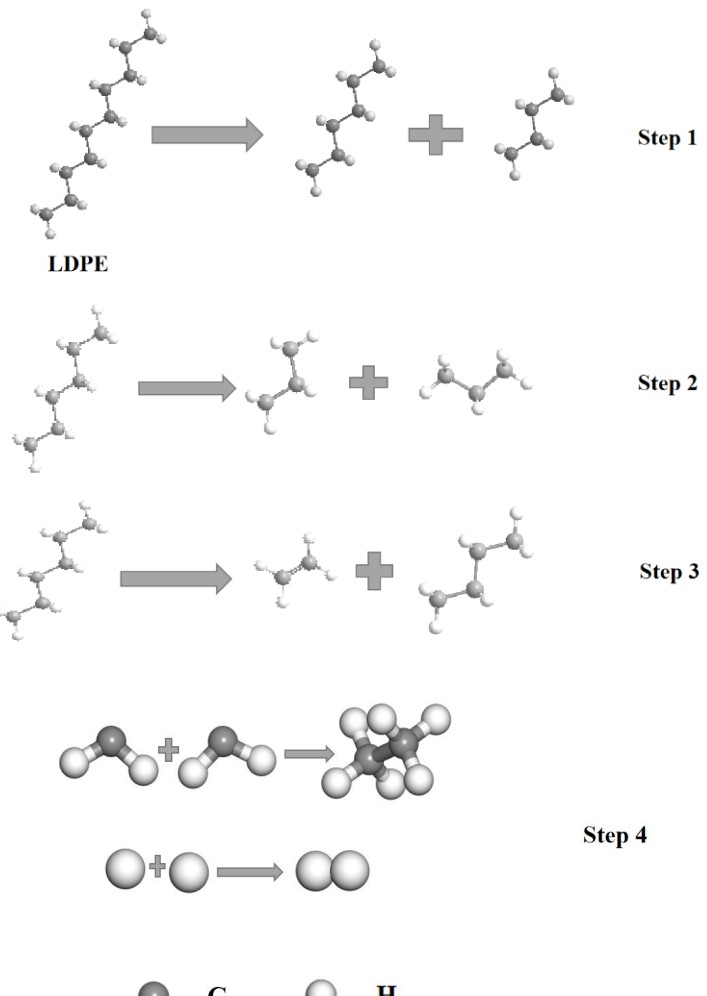

**Figure 6.** A plausible reaction pathway of LDPE pyrolysis without catalysts.

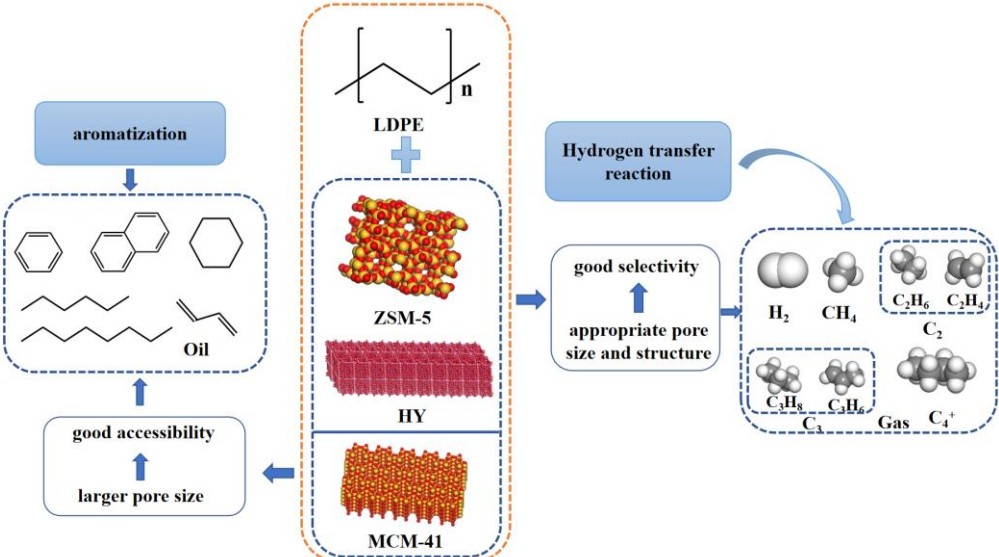

**Figure 7.** The scheme of catalyst participation in LPDE pyrolysis reaction pathways.

## 3. Materials and Methods

### 3.1. Materials

Powdered LDPE (100 mesh, Zhongyanshan Petrochemical Co., Ltd., Beijing, China) was commercially available. ZSM-5 powder with a $SiO_2/Al_2O_3$ ratio of 20, HY powder with a $SiO_2/Al_2O_3$ ratio of four and MCM-41 powder a with $SiO_2/Al_2O_3$ ratio of 30 were obtained from Zhongyanshan Petrochemical Co., Ltd. All of the molecular sieve catalysts were about 60–100 mesh in size and were calcined in air at 550 °C for 3 h before pyrolysis experiments.

### 3.2. Experimental Setup

The pyrolysis experiment of waste plastics was performed in a fixed bed reactor, as pictured in Figure 8. Briefly, the device was composed of an electric heating tube furnace, a temperature-controlled system, a quartz reactor (ID = 50 mm, L = 440 mm), as well as a cooling system. In the typical pyrolysis run, the catalyst and the plastic sample were mixed at a mass ratio of 1:2 (the proportion of catalyst to plastic remained constant throughout the experiment). Each plastic sample weighed approximately 5 g, the exact catalyst and plastic sample mixture were placed in the quartz tube between two sections of quartz wool. The tubular furnace was first blown with nitrogen (100 mL/min) for about 30 min to remove air and then the system was heated at a rate of 15 °C/min to the desired temperature (450, 500, 550, 600 or 650 °C) and held for 0 min.

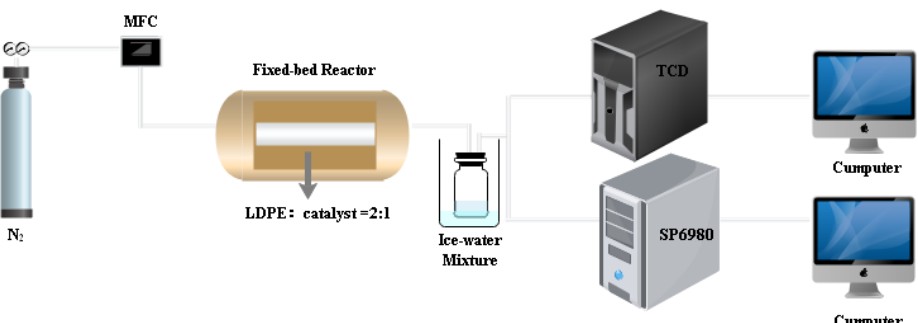

**Figure 8.** The pyrolysis experimental setup.

The gas and oil vapor generated from pyrolysis was blown into a condenser, which was cooled by the ice–water mixture. The condensate oil products were collected in a glass bottle and the mass of liquid oil was judged by the weight difference before and after the glass bottle. During the pyrolysis process, a solid mass was obtained by calculating the weight difference of the quartz tube before and after the reaction. The gas products were collected by gas bags and the mass was calculated by the difference. The equations involved were defined as follows:

$$Y_{p1} = \frac{m_1}{m_0} \times 100\% \tag{1}$$

$$Y_{p2} = \frac{m_2}{m_0} \times 100\% \tag{2}$$

$$Y_{p3} = \frac{m_0 - m_1 - m_2}{m_0} \times 100\% \tag{3}$$

where $Y_{P1}$, $Y_{P2}$, $Y_{P3}$ were the yields of oil, solid products and gas after pyrolysis, respectively. $m_0$ was the mass of the LDPE sample, $m_1$ and $m_2$ were defined as the mass of the liquid oil and solid product after pyrolysis.

Additionally, to ensure the accuracy of the experimental data, all experiments were repeated three times.

### 3.3. Characterization

The physicochemical properties of the three kinds of catalysts were determined by scanning electron microscopy (SEM), $N_2$ adsorption–desorption isotherms, temperature-programmed desorption of ammonia ($NH_3$-TPD) and high-resolution transmission electron microscopy (HRTEM). The detailed characterization methods of the samples are presented in the Supplementary Material.

### 3.4. Product Analysis

The determination of $H_2$ and $CH_4$ in cracking gas was done using GC-TCD (Ruihong, SP-6800A, Zaozhuang, China), analysis of hydrocarbon gases such as $CH_4$ and $C_2^+$ was done using GC-FID (Fuli, SP-6890, Nanjing, China). Each gas sample was measured three times to obtain the average. An analysis of pyrolysis oil composition was done using GC-MS (Agilent, 6890-5973, Santa Clara, CA, USA) with the HP-5MS capillary column (30 m $\times$ 250 μm $\times$ 0.25 μm). The operating parameters of GC-MS were described below: 60 °C for 3 min; 60 to 240 °C for 2 min at 12 °C/min; 240 to 300 °C for 10 min at 6 °C/min. The split ratio was kept at 100:1.

### 4. Conclusions

In this study, three different catalysts of ZSM-5, HY and MCM-41 were added, respectively, into non-catalytic pyrolysis of LDPE for regulatable oil and gas products. On the basis of analyzing the structure and characterization of the catalysts, the distribution of the pyrolysis products and the reaction mechanism of LDPE on different catalysts were discussed. The $NH_3$-TPD and BET characterizations of these catalysts exhibited the differences in pore size as well as acidity and their unique structural characteristics. The results of $NH_3$-TPD and BET presented that MCM-41 had the lowest acid strength and the largest pore size. The morphologies of the different catalysts were characterized by SEM and TEM. In the presence of MCM-41, a uniformly distributed granular structure could be observed. Because of the proper combination of acidity and structural properties, MCM-41 has been observed to produce a great deal of oil products, while ZSM-5 and HY were found to produce a great amount of gas products. Specially, ZSM-5 showed the greatest amounts of the aromatic products. This facilitates the selection of catalysts for cleaning applications of waste plastics and targeted access to valuable chemicals.

**Supplementary Materials:** The following supporting information can be downloaded at: https://www.mdpi.com/article/10.3390/catal13020382/s1. Table S1. Liquid product composition of LDPE at different temperatures. Table S2. Liquid product composition of LDPE at different temperatures. Table S3. Liquid phase GC-MS table of catalytic pyrolysis of LDPE by ZSM-5 molecular sieve at 500 °C. Table S4. Liquid phase GC-MS table of catalytic pyrolysis of LDPE by HY molecular sieve at 500 °C. Table S5. Liquid phase GC-MS table of catalytic pyrolysis of LDPE by MCM-41 molecular sieve at 500 °C.

**Author Contributions:** Resources, S.D.; data curation, Y.Z.; writing: original draft preparation, Y.L.; writing: review and editing, T.L.; project administration, H.Z. All authors have read and agreed to the published version of the manuscript.

**Funding:** This research was funded by the National Natural Science Foundation of China (22078168, NO. 52272086), Huawei Zhang, School of Environmental and Municipal Engineering, Qingdao University of Technology, Qingdao 266033, China.

**Acknowledgments:** Financial support was sponsored by the National Natural Science Foundation of China (22078168, 52272086).

**Conflicts of Interest:** The authors declare no conflict of interest.

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
