# Peer review of "Efficient Pyrolysis of Low-Density Polyethylene for Regulatable Oil and Gas Products by ZSM-5, HY and MCM-41 Catalysts"

_catalysts, doi:10.3390/catal13020382_

Round 1
Reviewer 1 Report
The manuscript entitled “Efficient pyrolysis of low-density polyethylene for regulatable oil and gas products by ZSM-5, HY and MCM-41 catalysts” evaluated the effect of these three types of catalysts on the composition and quantity of pyrolysis products in gas and oil phases. The ZSM-5 and HY catalysts produced a major amount of gas products and a higher content of aromatic compounds in the oil phase than uncatalyzed pyrolysis and pyrolysis catalyzed by MCM-41. Meanwhile, this latter catalyst exhibited the greatest oil yield with a high proportion of the C12–C18 fraction. In addition, a correlation between the catalytic results and acid properties and shape selectivity of catalysts was carried out. Although the study exhibited rigorously, I did not find significant novelty. These catalysts have been sufficiently studied in this type of process. The manuscript requires a thorough revision before being considered for publication in this journal.
General and Specific Comments
· In the abstract, the authors say: (a) “The addition of MCM-41 improved the oil yield, indicating that the secondary cracking of intermediate species in primary pyrolysis decreased with the case of catalysts” This only occurred on this catalyst. Revise the phrase. (b) “The excellent catalytic performance of MCM-41 may be attributed to its moderate total acidity and relatively high BET surface area.” The sentence: “excellent catalytic performance”, concerns which parameter?
· The paragraph on line 51 is repeated
· In line 71, instead isomerization should be isomerization capacity
· These types of catalysts in the pyrolysis of waste plastics have been extensively evaluated and reported. The authors should clearly indicate the novelty of this study.
· In the caption of figure 2, it must be specified which corresponds to a1, b1, c1, a2, b2 and c2
· In figure 3a, between 450 and 500 °C there is a sharp change in the amount of gas and liquid formed, and a plateau is reached. What causes this behavior? Do you have a thermogram of the polymer decomposition? This should be added to the manuscript
· In line 166 the authors say, “The HY catalyst clearly provides a higher gas yield as a result of the strong acid sites and high acid density of the HY zeolite” However, I don’t observe a significant difference between ZSM-5 and HY catalysts. Can you add the error bars to corroborate your statement? The trend in gas production between these two catalysts seems to be mainly related to the density and strength of the acid sites. The result of the MCM-41 catalyst confirmed this.
· Figure 4: All catalyzed reactions showed a percentage of C4+ around 95%, and for MCM-41 from 500 °C. Therefore no clear effect of the catalyst properties on this distribution was observed. However, the authors should expand their explanation.
· Line 198: There is an error
· In line 209, the authors affirm, “Moreover, it is obvious that the representative of diesel products is C12 ~ C18 hydrocarbons, and the three kinds of catalysts have potential application value in the use of plastic waste to produce diesel. Is it correct, considering that C12 ~ C18 hydrocarbons were mainly observed using MCM-41?
· Section 3.4 does not correspond to the pyrolysis mechanism. It is about the effect of catalysts on some reaction pathways. The section should be further revised.
· Figure 7: this is not a cracking mechanism. This is a scheme of catalyst participation in LPDE pyrolysis reaction pathways. This scheme should be thoroughly revised. According to their results, MCM-41 promoted oil fraction to a greater extent, and it was less favorable for cracking and aromatization reactions, resulting in a higher C12-C18 content. Meanwhile, the ZSM-5 and HY catalysts favored the cracking reactions leading to more gaseous products. Additionally, there was a higher aromatics content in the liquid phase, mainly with the ZSM-5 catalyst, probably due to its high acidity and shape selectivity. The lower proportion of aromatics obtained with MCM-41 can be related to its lower acid strength.
· Line 242: There is an error
· In line 246 says: “When the ZSM-5 is introduced into the pyrolysis process, the hydrogen extraction from the LDPE molecular chain by organic positive ions causes the carbon chain of LDPE to break and form the molecular chain” To which organic positive ions do the authors refer?
· In line 248 appears “as shown in Figure 7” Revise if really scheme explained the sentence mentioned
· Line 253: There is an error
· In line 256: Which dehydrogenation active sites do you refer to?
· In line, 258 authors say, “This is followed by the ring-opening reaction of aromatic ring catalyzed by the ring-opening active site” What ring-opening active sites are you referring to?
· In line 268 appear: “It can be seen that the coke deposition on HY is highest among the three kinds of catalysts, which is due to the higher acid site and surface area of HY catalyst” Was this measured by the authors? If so, this should be included in the manuscript.
· Line 269: There is an error
· In experimental setup: What was the pyrolysis time? Was heating at 15°C/min carried out with the sample in the reactor?
· In the Characterization of sample: The NH3-TPD method for acidity determination should be included.
Reviewer 2 Report
Liu, Li and Co-Workers described a «DEfficient pyrolysis of low-density polyethylene for regulatable oil and gas products by ZSM-5, HY and MCM-41 catalysts». The article is well written, all methods are relevant and necessary to describe the results and conclusions obtained in this work.
While reading, there were a few questions and comments:
1) Based on the data presented in paragraph 4.2. somewhat unclear, the samples were heated at a rate of 15 °C/min to the set temperature and held for 0 minutes. After that, how the samples were unloaded and after what time, in order to clear out secondary reactions. How was the cooling carried out and how long was the reactor cooled down?
2) What were the solid products of pyrolysis? Based on Fig. 3, which shows that with increasing temperature, the yield of solid products decreases, it can be excluded that this is "coke". Please clarify this point.
3) Was a carbonaceous residue formed at high pyrolysis temperatures?
4) Please clarify why in the composition of gaseous products during pyrolysis in the presence of ZSM-5, where the yield of aromatic compounds is the highest, the hydrogen content is at the level of 0%?
Reviewer 3 Report
The experimental work has value and adds to the existing literature. Given the study title and scope, authors are advised to focus on the results of pyrolysis experiments rather than catalysts characterization. This article section could be inserted as supporting content. The results section is limited to catalyst characterization, and pyrolysis findings are not presented herein. The overall structure of the results and discussion sections is confusing. Authors are advised to combine these sections.
Please check the enclosed file, report the comments on a file, answer them and report the amended text.

Round 2
Reviewer 1 Report
n
Reviewer 3 Report
Authors addressed all comments. Sincerely.